# Nutritional Risks among Adolescent Athletes with Disordered Eating

**DOI:** 10.3390/children8080715

**Published:** 2021-08-21

**Authors:** Ioanna Kontele, Tonia Vassilakou

**Affiliations:** Department of Public Health Policy, School of Public Health, University of West Attica, Athens University Campus, 196 Alexandras Avenue, 11521 Athens, Greece; ioannakontele@gmail.com

**Keywords:** adolescent athletes, eating disorders, disordered eating, nutritional risk, low energy availability, female athlete triad

## Abstract

In their attempt to achieve the optimum weight or body shape for their activity, athletes frequently use harmful weight-control practices that may lead to the development of disordered eating or eating disorders. These practices are linked to several medical and mental consequences that may be more serious in adolescent athletes, as their bodies must meet both intensive growth demands and training requirements at the same time. Among other consequences, adolescent athletes may be at nutritional risk, due to their high nutrient needs and unhealthy eating behaviors. A literature review was conducted to examine the main nutritional risks and malnutrition issues faced by adolescent athletes that present disordered eating attitudes or eating disorders. Most studies refer to adult elite athletes, however research on adolescent athletes also indicates that the most common nutritional risks that may arise due to disordered eating include energy, macronutrient and micronutrient deficiencies, dehydration and electrolyte imbalances and changes in body composition that may lead to menstrual abnormalities, and decreased bone mass density. Educational programs and early detection of disordered eating and eating disorders are crucial to avoid the emergence and ensure timely management of nutrition-related problems in the vulnerable group of adolescent athletes.

## 1. Introduction

Adolescence is the second fastest growth period after infancy and is characterized by significant alterations in body composition, metabolic and hormonal function, organ’s maturation, and formation of nutrient deposits that may have an impact on future health [1]. Adolescence is also a challenging period for a person’s nutrition, as adolescents are exposed to different forms of malnutrition, such as undernutrition, obesity and deficiencies of certain nutrients [2]. Moreover, adolescence is a vital time for establishing eating habits that will likely last during a person’s life [3]. Even if nutritional vulnerability may not be as great as in infancy and childhood, adolescents are a nutritionally susceptible group of population, because of their high growth requirements, unhealthy eating patterns and high-risk behaviors [2]. Organized sports provide many physical, mental and cognitive benefits to young people, however adolescent athletes have very high nutritional requirements, as a result of intensive daily training in addition to their energy and nutrients needs for growth and development [4]. Adolescents who participate in sports have higher nutritional demands than their non-athletes peers of the same age, in order to meet at the same time their needs for growth and wellness, as well as for optimal sports performance [4,5,6,7,8,9].

Despite their high nutritional requirements, athletes often engage in inappropriate dietary strategies in their attempt to regulate their body weight and/or body shape, for the purpose of reaching optimal sports performance or ideal physical standards for their sport [8,10,11,12].

Athletes are at a higher risk of developing disordered eating than a clinical eating disorder [10,11]. Thus, they may use unhealthy weight-control practices, such as skipping meals, restrictive eating, over-exercising, dehydration techniques, vomiting, using diuretics, laxatives, or diet medications, but without fully meeting the criteria for eating disorders [8], as they are described in DSM-5 [13]. According to Sundgot-Borgen et al. (2013) and Wells et al. (2020), disordered eating behavior develops in an athlete across a spectrum. This spectrum starts from optimal nutrition (that includes healthy dieting that leads to gradual weight loss), continues to the occasional use of more intensive weight-loss methods such as short-term restrictive diets and may progress to more extreme behaviors, such as chronic energy restriction, eating nothing for one or more days (fasting), using passive or active dehydration techniques, self-induced vomiting, frequent use of diuretics, laxatives, or diet medications, and excessive exercise. This sequence may end with the presence of clinical eating disorders that fulfill the criteria of DSM-5 [8,14,15]. According to Wells et al. (2020), “individual athletes can move back and forth along the spectrum of eating behavior at any point in time over their career and within different stages of a training cycle” [8].

Disordered eating prevalence is higher among adolescent elite athletes compared to their non-athlete peers and it may affect athletes of any gender, sport type or level of competition [16,17]. It is estimated that up to 45% of adolescent athletes develop disordered eating attitudes, especially females in aesthetic and weight-sensitive sports [16,18,19,20,21,22,23,24,25,26,27,28,29,30,31,32,33]. Disordered eating etiology is multi-factorial, as socio-cultural, psychological, family, biological and genetic factors interrelate [14]. Moreover, in athletic population, risk factors that are also associated with the development of disordered eating are high competitiveness, starting sport-specific training in young age, weight class rules and pressures from the sport environment to attain a certain body weight or shape [14]. Weight pressures in the sports environment include the comments from coaches, judges or teammates regarding weight and body shape, requirements for specific weight or body shape to compete, the revealing nature of the training or competition uniform and the perception that low weight gives performance advantages [34,35]. Although athletes from any sport may experience the above mentioned pressures, it seems that athletes from aesthetic and weight-class sports experience higher risk to develop disordered eating due to weight pressures [33,34,36]. Regardless of the severity of disordered eating, health and performance consequences are serious and they increase as disordered eating attitudes turn into a clinical eating disorder [8]. The health effects of disordered eating depend on various factors, such as the person’s age, other health problems and body composition prior to the weight loss initiation, as well as the duration, volume and rate of energy restriction and purging behaviors [14,25]. Physiological and medical complications affect most of the body systems, including the cardiovascular, gastrointestinal, endocrine, skeletal, renal, reproductive, and central nervous systems, and may also be linked to psychological stress and depression [16,37,38]. Disordered eating attitudes may also cause negative effects on sport performance due to low energy availability, excessive body fat and lean mass loss, fluids’ and electrolytes’ imbalance, that may increase illness and injury risk, cause limitations in quality and consistency of training and impair recovery after training or injuries [39,40,41,42]. These effects depend also on the duration and the severity of disordered eating [43]. Long-term consequences are also possible, but the research regarding the possible damages on health, that may last or arise many years after the initiation of disordered eating behaviors, is very scarce. One possible consequence is the long-term maintenance of the disorder. Sundgot-Borgen et al. (2012) reported that just 72% of former elite athletes with eating disorders during their athletic career had recovered after 15–20 years [44]. Moreover, athletes with long-term inadequate energy and nutrient intake as well as menstrual disturbances during the growth period of adolescence, may never achieve an optimal peak bone mass, resulting in greater risk of development of osteopenia or osteoporosis later in life [45]. Growth retardation is also possible but studies on elite young athletes show a catch-up effect for bone and body mass growth when energy intake becomes normal [46,47]. Finally, there is the hypothesis of greater cardiovascular disease risk later in life, due to endothelial dysfunction, which has been found in female athletes suffering from amenorrhea and energy deficiency [48,49,50].

During the past two decades, there has been a plethora of published studies that have examined the potential harmful consequences of disordered eating attitudes among athletes. However, research has focused primarily on adult (mainly college-age) elite female athletes [11,51,52,53,54,55,56]. Less studies have examined the health consequences, and especially those related to the nutritional status, of disordered eating behaviors in adolescent elite and non-elite athletes. The present article aims to review the existing literature on the common nutritional risks and malnutrition issues faced by adolescent athletes that present disordered eating or eating disorders and, to emphasize the value of nutritional interventions regarding the prevention and early treatment of disordered eating in adolescents participating in sports.

## 2. Materials and Methods

An electronic search of the international literature was carried out, using three different databases (PubMeD/MEDLINE^®^ (United States National Library of Medicine (NLM), Bethesda, MD, USA), PsycInfo (American Psychological Association, Washington, DC, USA) and Google Scholar (Google, Mountain View, CA, USA) in order to track relevant observational studies, reviews, systematic reviews and meta-analyses published until May 2021. The keywords that were used in our search include the following: “adolescent athlete”, “high school athlete”, “nutritional risk”, “malnutrition”, “undernutrition”, “nutritional deficiencies”, “low energy availability”, “dehydration”, “disordered eating”, “eating disorder”, “female athlete triad”. All articles were reviewed for relevance. Included studies had to provide data about the correlation between disordered eating or eating disorders and specific nutritional risks.

For the scope of our review, “adolescents” were considered males and females aged 10 to 19 years, according to WHO definition [2]. Under this condition, articles that defined in detail the age of the studied population or mentioned that the sample consisted of high-school athletes were included. Studies involving only adult athletes (>19 years of age) and college-age athletes were excluded, as well as studies not declaring age groups. Moreover, a small number of studies included both adolescents and adults, but our review presents only the results regarding adolescents.

Moreover, the review includes articles regarding adolescents that participate in various organized sports in elite or non-elite level. Articles regarding participation in recreational activities have been excluded. Articles of adolescents who did not systematically participate in sports were also excluded.

Finally, articles that were written in non-English languages and those that had no full text available were excluded. There was no restriction regarding the year of publication.

## 3. Results

Adolescent athletes are more vulnerable than adult athletes to the physical consequences of disordered eating attitudes, due to their high energy and nutrient requirements [4,5]. Their sport performance is compromised, but also normal growth, development and maturation may be impaired [5,40,42].

Nutritional risks that may arise due to disordered eating include energy, macronutrient and micronutrient deficiencies, dehydration, electrolyte imbalances, menstrual irregularities, and decreased bone mass density [37,38,40]. Moreover, disordered eating is associated with the development of the “Female Athlete Triad” and the “Relative Energy Deficiency in Sport Syndrome (RED-S)” [41,42].

The present electronic search retrieved 463 articles. After thorough examination of article abstracts and full-texts, a final number of 34 articles considered eligible to be included in the present review. Twenty-eight of them were articles of original researches, four were reviews and two were clinical reports. The majority (19 articles) examined mainly the association between disordered eating and “Female Athlete Triad” or bone mineral density, 15 articles discussed “low energy availability”, three articles examined nutrient deficiencies of athletes with disordered eating and five articles presented the topic of dehydration and electrolyte imbalances due to disordered eating. Some articles examined more than one of the above-mentioned issues.

### 3.1. Low Energy Availability

“Energy Availability” is the amount of dietary energy available for other body functions after exercise training, and it is calculated by subtracting the energy expended during exercise from the dietary energy intake [57]. Energy availability is usually adjusted to fat-free mass (FFM), since the majority of energy is spent by FFM [57]. Energy availability for optimal bodily function is normally 45 kcal per kg of FFM per day [57], but in adolescents, who are still growing and developing, it may be much higher [48]. Low energy availability (LEA) refers to an imbalance between the energy intake that a person ingests from foods compared to the energy expenditure for exercise, allowing inadequate energy quantity to support the necessary human physiological functions required to maintain optimal health [42]. Although there is no specific cut-off point for energy that defines LEA, it seems that in the case that energy availability falls below 30 kcal/kg FFM/day, the human body tends to reestablish energy balance and prolong survival, but reproductive and skeletal health are impaired [57,58,59]. Moreover, energy availability of 30 kcal/kg FFM/day is just below the average resting metabolic rate (RMR) of healthy adults [57].

Athletes are often in LEA status due to increased energy expenditure for exercise or due to low energy intake from food that is not enough to cover their high exercise energy demands [58]. LEA may occur unintentionally due to lack of knowledge regarding the best practices to balance the amount of daily energy needed for training and normal physiological functions by the athletes [60]. However, LEA may also occur due to intentional decreased energy intake as noticed in the cases of disordered eating, such as food restriction, vomiting, stimulants or laxatives use, in an attempt to control body weight [61]. 

Low energy availability (whether intentional or unintentional) may cause health impairment and lead to several medical complications, including problems in the function of cardiovascular, endocrine, reproductive, and skeletal systems, since, as a response to LEA, physiological processes lower the amount of energy utilized for cellular maintenance, thermoregulation, growth, and reproductive function [37,62]. LEA is also a common cause of menstrual dysregulation and amenorrhea in adolescent female athletes [63,64], and it is considered the underlying cause of the “Female Athlete Triad” [37,48] as well as the “Relative Energy Deficiency in Sport Syndrome” (RED-S) [41]. Health effects of LEA, especially amenorrhea and imbalance in bone remodeling, are critical for female adolescent athletes, because the development and maturation of the reproductive system, and the attainment of peak bone mass density is achieved during this period [14]. “Female Athlete Triad” and low bone mineral density are further discussed in Section 3.3.

Many surveys have investigated the prevalence and consequences of LEA in adolescent athletes. A study of 36 elite competitive adolescent female figure skaters in the USA found that 25% of the athletes scored above cut-off score in disordered eating screening test (EAT-40) and reported lower than recommended energy intakes [65]. In a study in Brazil, 91.7% of adolescent female tennis players fulfilled the criteria for disordered eating and/or low energy availability. The average energy availability of the athletes was 31.17 kcal/kg FFM/day, while 87.5% were found to have energy availability less than 45 kcal/kg FFM/day and 33.3% below 30 kcal/kg FFM/day [66]. In another study of 77 adolescent female swimmers also in Brazil, all the athletes who consumed less than 30 kcal/kg FFM/day presented disordered eating behaviors [67]. Moreover, Wood et al. (2021) found that female adolescent endurance runners with a high level of cognitive dietary restraint reported lower energy consumption than runners without dietary restraint [68]. Low energy availability has also been reported in 36% of female high-school athletes of various sports [69], in adolescent female acrobatic gymnasts, where the average energy intake was 32.8 ± 9.4 kcal/kg FFM/day [70], in female adolescent elite football players in Germany, with 53% of them reporting energy availability of less than 30 kcal/kg lean body mass/day [71] and in female and male high-school cross-country runners in the USA, with 30% and 60% respectively meeting the criteria for LEA [72]. It is worth to mention that, although adolescent female athletes may be considered at a greater risk of inadequate energy intake [67] and the research investigating adolescent male athletes is relatively sparse, studies of male adolescents have demonstrated that they are also at risk for disordered eating and low energy availability [42,73,74].

Although athletes lower their energy intake to achieve weight loss, LEA has been correlated with higher body fat percentage and decreased resting metabolic rate (RMR) [75,76]. In a study of 353 young athletes in Germany, female athletes with LEA had higher body fat mass than females with normal energy availability [75], while a study of 42 gymnasts in the USA showed that energy deficits are associated with higher body fat percentage [76]. These findings are possibly explained due to an adaptive reduction in RMR of the athletes with low energy intake [76], and higher leptin concentrations [75]. LEA has also been associated with decreased RMR in studies among adult male and female athletes [77,78].

In conclusion, it seems that LEA is frequent among adolescent male and female athletes in various sports and that it is often correlated to disordered eating behaviors.

### 3.2. Macronutrient and Micronutrient Deficiencies

Low energy availability is often accompanied by low macronutrient and micronutrient intakes [58]. In athletes, reduced intakes of macronutrients may cause decrease of the physiological ability for bone formation, muscle mass maintenance, damaged tissue repair, and injury recovery [58]. Glycogen stores may not be properly replaced during time periods of intensive activity training if carbohydrate intake is inadequate [79]. At the same time, protein requirements may rise, because body protein stores may be used as alternative energy source [58]. Micronutrients are also required for the formation of bones and muscular tissue, the replenishment of red blood cells, and the provision of co-factors for the regulation of energy-producing metabolic pathways [58]. Athletes who consume less than required calories or restrict certain food groups from their diet, in an attempt to control their body weight, may undergo low nutrient intakes, which may result to ineffective use of energy intake regarding health maintenance and provision of adequate energy for physical activity [58,80,81].

Studies that have examined nutrient intakes of adolescent athletes with disordered eating are very scarce. Costa et al. (2013) found that adolescent athletes with disordered eating behaviors presented lower protein and calcium intakes, while 70% of them had carbohydrate intake below the recommended levels [67]. Wood et al. (2021) found that female adolescent endurance runners that intentionally restricted their dietary intakes had lower carbohydrate intake than runners with normal eating attitudes [68]. Higher scores in the Eating Attitudes Test 26 (EAT26 screening tool) were also associated with lower micronutrient intakes in a study of 21 adolescent competitive female figure skaters [82].

Even though data on nutrient deficiencies in this population group are limited, it is expected that athletes who restrict their energy intake or specific food groups, as well as those that use purging behaviors, will be deficient in a number of essential nutrients. 

### 3.3. Low Bone Mass Density

Adolescence is a critical period for bone growth, acquisition and maturation, as the greatest rate of bone mass formation occurs during puberty. Almost 90% of peak bone mass density (BMD) is achieved by 18 years of age, and 25% of bone mass is formed during the two years around menarche [83]. Bone mass that is formed during the years of childhood and adolescence is important for the achievement of peak bone mass density of the person and the prevention of osteoporosis later in adulthood [84]. Achievement of optimal BMD during adolescence depends on adequate nutrition, including overall adequate energy availability, intakes of certain nutrients (especially vitamin D and calcium), and regular weight-bearing physical activity [48,62]. Increased risk of low BMD has been linked to several factors, such as low body weight, over-exercise, dietary restrictions and menstrual irregularities (late menarche, oligomenorrhea, amenorrhea) [83,85].

In athletic populations, menstrual abnormalities and low bone mass density are associated to low energy availability as a result of decreased energy intake or over-exercise [37]. Energy intake is considered a more crucial factor for the formation of bone mass than certain nutrients (calcium and vitamin D), since energy restriction causes hormonal profile changes resulting to decreased calcium absorption and increased calcium mobilization in the bones [86]. 

Athletes with disordered eating behaviors may experience low energy availability, which disrupts the normal cycle of menstruation, and, eventually, results to imbalance in bone remodeling that may lead to osteopenia or osteoporosis [37]. The three interrelated conditions of amenorrhea, osteoporosis, and disordered eating have been recognized as the syndrome of “Female Athlete Triad”, which was first described in 1992 [37,40]. In 2007, the American College of Sports Medicine revised its position statement regarding the triad suggesting that each one of the three clinical conditions comprises the pathological end of a spectrum of interrelated, subclinical conditions between health and disease. Therefore the three conditions were renamed to (a) menstrual function, (b) bone mineral density, and (c) energy availability, to more appropriately depict the entire spectrum, which may vary from excellent health to sickness in each component [37]. Each one of the triad components may exist independently, but it seems that the emphasis on weight loss and low energy intake may initiate a cycle where all three diseases occur in sequence [43]. “Female Athlete Triad” is particularly harmful during adolescence, as the achievement of maximum bone mass density, as well as the development and maturation of the reproductive system, take place at this age [14,48].

The “Female Athlete Triad” syndrome affects female athletes of any sport type and competition level [62]. The prevalence of the full syndrome in adolescent female athletes seems to be relatively low and ranges from 1.2% to 4.2% [20,24,66]. Yet, a much greater percentage of female adolescent athletes may present with 1 or 2 components of the triad [19,20,24,60,66,69,87]. In studies of adolescent athletes, prevalence of disordered eating ranges from 18.2% to 91.7%, prevalence of menstrual irregularities is found between 18.8% and 48.0%, and prevalence of low bone mass ranges between 15.4% to 42.1% [19,20,24,60,66,69,87].

The assessment of menstrual abnormalities prevalence among adolescents involves a number of challenges, as it is common for girls during their first year after menarche to experience amenorrhea and oligomenorrhea [88]. Yet, the results of studies among adolescent athletes show that menstrual irregularities are quite common in this group and are linked to disordered eating and low energy availability [19,20,25,60,89,90,91]. Moreover, several studies in adolescent athletes have found a correlation between low bone mass and both disordered eating and menstrual dysfunction [20,25,68,90,91,92,93,94]. 

Weimann et al. (2000) studied 22 female adolescent elite gymnasts in Germany and found that female gymnasts experienced delayed menarche, bone formation retardation, and reduced height potential, while their nutritional intake was insufficient [89]. Beals (2002) investigated the nutritional status, eating behavior and menstruation of 23 nationally ranked female adolescent volleyball players. Their average caloric intake was lower than their average calorie expenditure, while 17% reported past or present amenorrhea, 13% past or present oligomenorrhea and 48% irregular menstrual cycles. Almost half of the athletes reported actively “dieting” [60]. Nichols et al. (2007) examined the association of disordered eating and menstruation irregularities among 423 high-school athletes and found that athletes with oligomenorrhea and amenorrhea consistently reported higher levels of dietary restriction and higher scores on disordered eating scales (EDE-Q) than eumenorrheic athletes, while athletes with disordered eating were more than twice as likely to report oligo/amenorrhea than athletes without disordered eating behavior [19]. In another study of 170 female athletes, Nichols et al. (2006) found that athletes with oligomenorrhea and amenorrhea had significantly lower trochanter BMD values compared to athletes with regular menses. Moreover, among athletes with disordered eating, those who reported pathological behaviors had lower BMD for all bone sites compared to those who reported normal eating behaviors [20]. Barrack et al. (2010) found that female adolescent runners whose bone turnover was elevated had lower body mass index, less menstrual cycles during the past year, and lower than the recommended energy intakes compared to those with normal bone turnover [92]. Thein-Nissenbaum et al. (2011) studied 311 female high-school athletes and found that athletes reporting disordered eating attitudes were twice more likely to suffer an injury and to experience menstrual irregularities during the season than those who reported normal eating behavior. Moreover, aesthetic and endurance sports athletes who reported disordered eating were eight and three times, respectively, more likely to suffer an injury than athletes from the same sports who reported normal eating behaviors [25]. These findings were similar with the results reported by Rauh et al. (2010), who found that female athletes from 6 high schools in California reporting disordered eating were nearly three times more likely to experience menstrual disorders than their counterparts who reported normal eating behaviors. Moreover, disordered eating, menstrual irregularities and low BMD were linked to musculoskeletal injuries [90]. A study on female and male adolescent runners by Tenforde et al. (2015) found that the main risk factors for decreased BMD Z-scores in girls were current menstrual irregularities, menarche at a later age, lower fat free mass, and lower milk consumption. In boys, lower BMD Z-scores were associated with lower body mass index (BMI) Z-scores, as well as with the belief that thinness improves performance in sports [93]. In a study of 320 adolescent female athletes, Thralls et al. (2016) found that underweight was linked to a greater likelihood of menstrual abnormalities and low BMD. In fact, athletes whose BMI was lower than the 5th percentile for their age were nine times more likely to report menstrual abnormalities and had lower BMD compared to those whose BMI was between the 50th and 85th percentile [91]. A study of 390 elite female athletes in Japan found that among adolescent female athletes, the risk of stress fractures was increased by 12.9 times in those with secondary amenorrhea, 4.5 times in those with low BMD for the whole body, and by 1.1 times in those with a low ratio of actual body weight to ideal body weight [94]. Finally, a recent study on 40 adolescent female endurance runners in the USA found that the athletes with elevated cognitive dietary restriction exhibited significantly lower BMD Z-scores in lumbar spine compared to the athletes with normal eating behavior [68].

It seems that there is substantial amount of scientific evidence regarding bone mineral density imbalances in athletes. It can be assumed that low bone mineral density in adolescent athletes is correlated in a direct way to disordered eating attitudes (such as low consumption of energy or certain nutrients) as well as in an indirect way as a result of disordered eating consequences, such as low energy availability and menstrual irregularities.

### 3.4. Dehydration

Athletes in weight-sensitive sports often use hypohydration and dehydration techniques in an attempt to lose weight and, thus, obtain a perceived advantage in a competition or compete in a lower weight category [95]. Common dehydration practices used by athletes include restriction of fluid intake, spitting, vomiting, use of diuretics and/or laxatives, steam baths, saunas, and wearing nonporous suits to increase sweat production [14,96].

It is estimated that up to 67% of athletes participating in weight-class sports (e.g., Taekwondo, wrestling, boxing) try to lose weight with various dehydration practices [97]. Most studies focus on adult athletes. In a study of 2532 high-school wrestlers, 2% reported weekly use of laxatives, diet pills, or diuretics, while the same percentage reported at least weekly self-induced vomiting to lose weight. Fasting and various dehydration methods were the primary techniques used for rapid weight loss [98]. In a study of 4746 adolescent athletes it was found that, compared to athletes of non-weight-related sports, males participating in a weight-related sport were more likely to report vomiting, laxative and diuretic use during past week and past year, while females participating in a weight-related sport were more likely to report vomiting and laxative use during past week and past year [74]. A more recent study of 1138 elite adolescent athletes in Germany indicated that passive or active dehydration practices (e.g., sauna, exercise in sweatsuits) were the most commonly used methods to control weight [33]. The ATHENA program, that was conducted in a sample of 1668 female team sport athletes of 18 high schools in the USA found that 4% of the athletes had used vomiting, 1% diuretics and 1% laxatives during the past 3 months, while 14% had not eaten for one or more days, in order to cause rapid weight loss [22].

Maintaining adequate fluid balance during exercise is crucial for dehydration prevention and maintenance of normal cardiovascular and thermoregulatory functions required for effective exercise performance [99]. Sport performance and health may be affected by hydration status, which refers to body water content as well as to body electrolytes concentration [100]. A dehydrated person’s lower blood volume may compromise thermoregulatory capacity during exercise, resulting in poor performance [100,101]. With high-intensity exercise, dehydration causes decreased strength, power, and endurance [102,103,104] as well as increased body temperature and higher susceptibility to heat illness [96]. Moreover, electrolyte concentration changes caused by dehydration are likely to be detrimental to muscular function, resulting in muscle mass loss and decreased strength and power [101]. Hypohydration has an impact on muscle metabolism by hastening glycogen depletion, and central nervous system function by lowering motivation and effort [100]. Therefore, athletes who practice severe energy or fluid restriction to lose weight may suffer unfavorable repercussions such as loss of lean tissue, hormonal disturbances, and performance impairment [95,101].

In adults, dehydration of at least 2% of body weight has been linked to a reduction in endurance and work capacity [99]. It is unclear how much dehydration impacts adolescent endurance performance [99]. Unlike adults, we still do not know the hypohydration level that is associated with adverse athletic performance and negative health effects in young athletes [100].

Dehydration and electrolyte imbalances have been thoroughly studied in sports, as they result in serious health and performance problems. Unfortunately, it seems that adolescent athletes with disordered eating often use harmful weight control techniques that cause detrimental imbalances regarding hydration status.

## 4. Discussion

The scope of the present review is to present the available literature evidence regarding the main nutritional risks and challenges faced by adolescent athletes who adopt disordered eating behaviors in their attempt to control their body weight, aiming to meet the ideal physical standards for their sport. During the last 20 years a great deal of research has been implemented regarding the health and performance consequences of disordered eating and eating disorders in adult athletes. This work has resulted in a number of guidelines and position papers, such as the International Olympic Committee (IOC) consensus statements on Relative Energy Deficiency in Sport [41,42]. The amount of research regarding the vulnerable group of adolescent athletes is limited, and it is focused mainly on the prevalence of disordered eating as well as the prevalence of the “Female Athlete Triad”.

As it was mentioned earlier, athletes have a higher risk of developing disordered eating than a clinical eating disorder [10,11]. Moreover, clinical interviews are necessary to examine the prevalence of clinical eating disorders, according to the specific diagnostic criteria [13]. Most studies on adolescents use questionnaires that can assess symptoms associated with eating disorders rather than clinical interviews [16]. This is the reason that the majority of studies presented in this review discuss the associations between disordered eating behaviors (and not eating disorders) and their consequences.

According to the present literature review, the most well documented consequences of disordered eating regarding the nutritional status of adolescent athletes are low energy availability and low bone mass density. Nutrient deficiencies and fluids’ and electrolytes’ imbalances due to disordered eating have also been studied to a lesser extent.

Low energy availability is well documented in studies of adolescents, as well as adult athletic populations. The prevalence of disordered eating is considered to be higher among athletes who participate in weight-sensitive sports, such as aesthetic and endurance sports, as well as in weight-class sports [11,18,19,25,33], but some studies in adolescent athletes show that athletes from non-weight-sensitive sports may also use pathogenic weight control behaviors [16,22]. Respectively, low energy availability is also present in adolescent athletes of weigh-sensitive sports [65,70], as well as of non-weight-sensitive sports [66,71].

Studies from various countries and different sports agree that adolescent athletes, who report disordered eating attitudes, consume less than the required energy through their nutrition plan [65,66] and that athletes that choose unhealthy ways to control their weight have lower energy availability than athletes who don’t engage in unhealthy behaviors [67,68]. Using the commonly agreed limit of 30 kcal/kg FFM/day, studies have found that 30% to 60% of athletes experience low energy availability [66,69,70,71,72]. It seems that a significant number of athletes do not consume enough energy, necessary not only to cover their training demands, but most importantly for the maintenance of their normal physiological functions. This situation may cause medical complications in several body systems and impair growth and maturation [10,37,62]. There is a great amount of research evidence regarding health consequences of LEA, especially in adult female athletes. The most predominant complications concern the function of endocrine system, including the disruption of the hypothalamic-pituitary-gonadal axis that causes functional hypothalamic amenorrhea [42]. Moreover, endocrine system is affected regarding thyroid function and secretion of various hormones, such as insulin, cortisol, growth hormone and appetite-regulating hormones (leptin, ghrelin, adiponectin and peptide YY) [42,59,63,105,106]. It has also been hypothesized that functional hypothalamic amenorrhea and the alterations in hormone secretion may also cause detrimental consequences on future pregnancies [105,107]. LEA has also been associated with increased susceptibility to upper respiratory tract and gastrointestinal tract infections [108].

The research regarding LEA in male athletes is limited and it appears that men’s physiology is characterized by a greater resilience in LEA regarding the effects on endocrine system and bone metabolism [109,110], probably because women’s energy demands are associated to gestation. Nevertheless, there are studies suggesting that adult male athletes may also develop suppression of the reproductive function known as exercise hypogonadal male condition [105]. Male athletes in LEA state are also expected to have reduced testosterone [42,93], leptin and insulin [110] as well as reduced skeletal muscle protein and muscle glycogen stores [109,111,112]. Another significant finding is that male athletes who restrict their energy intake have negative psychological effects and high risk of bulimic symptomatology [111,113]. A less studied effect of disordered eating in sports is the “result” of these methods regarding the desired reduction in body weight and fat percentage of the athletes. Two studies demonstrated that low energy availability was correlated with higher body fat percentage and decreased resting metabolic rate [75,76]. Metabolic adaptations that are physiologically expected to occur after a period of low energy availability will probably cause a decline in BMR and a weight loss plateau, that may lead the athlete in further caloric restriction to continue losing weight. This vicious cycle may continue and lead to long-term state of low energy availability, as well as to the development of an eating disorder [114]. Athlete’s RMR and body composition may be further negatively affected by macronutrients and micronutrients deficiencies. Adolescent athletes that use unhealthy weight control methods have lower than recommended intakes of carbohydrates, protein, calcium, and iron [67,68,82]. Moreover, low energy intake is expected to be associated with low intakes of other nutrients too [58]. These nutrient deficiencies, along with athlete’s increased requirements due to growth and training, may cause physiological inefficiency to repair muscles after training or create new muscle mass effectively [58]. Muscle mass may be further depleted when glycogen stores are exhausted due to low carbohydrate intake [58,79]. These findings suggest that athletes’ efforts to control their body weight by unhealthy methods do not provide athletic performance benefits. Moreover, they may even result to the opposite than desired results, leading to muscle mass decrease and body fat increase. Therefore, athletes should be discouraged from using unhealthy weight-loss methods in order to achieve the desired body composition.

Furthermore, nutrient deficiencies along with dehydration and electrolyte imbalances are detrimental to athletic performance [4,7,100]. In both children and adults, maintaining fluids’ and electrolytes’ balance is critical for athletic performance and thermoregulation [115], as dehydration causes abnormalities in cardiovascular and thermoregulatory functions, decreases in strength, power and endurance and raised susceptibility to heat illness [96,99,101,102,103,104]. Unfortunately, dehydration techniques such as use of diuretics and laxatives, vomiting, steam baths, saunas and exercise in sweatsuits are used by a large number of athletes to lose weight. However, the number of studies investigating this topic on adolescent athletes is small [22,33,74,98].

The “Female Athlete Triad”, that leads, as a sequence, from low energy availability to menstrual irregularities and, finally, to suboptimal bone mineral density is probably the most well documented consequence of disordered eating in sports [37,48,63,88]. Since the development of the reproductive system and the peak bone mass density achievement occur during adolescence, the consequences of the triad are particularly severe for female adolescent athletes [14,48,83,84]. Even in the case of normal menstruation, poor diet and low BMI are more likely to be associated with low bone mass density or musculoskeletal injuries than with normal diets [48]. Athletes that use unhealthy weight control methods are at increased risk to develop low bone mass density, as they usually face simultaneously a number of risk factors, such as low energy availability, menstrual irregularities, underweight and calcium deficiency [83,85].

The triad may affect female athletes from all sports and competition levels [62]. Athletes from weight sensitive sports may be at a greater risk for delayed menarche and bone maturation due to their disordered eating attitudes used to control their body weight [25,89], but adolescent athletes from non-weight-sensitive sports may also present menstrual abnormalities [60]. Many studies of female adolescent athletes of various levels demonstrate that disordered eating is associated with the development of menstrual problems, low bone mineral density and musculoskeletal injuries [19,20,25,68,90,91,92].

Although the “Female Athlete Triad” is a syndrome that concerns only females, there is growing evidence that male athletes (especially from weight-class sports), may also experience the underlying causes of low energy availability, as they engage in unhealthy weight control behaviors [42,73,74]. These practices also cause low bone mineral density in males [93], as long as other health and performance consequences of disordered eating [43,53,73]. In 2014, the “International Olympic Committee” (IOC), considering the fact that male athletes are also affected by disordered eating and low energy availability, introduced the term of “Relative Energy Deficiency in Sport” (RED-S) [41]. The RED-S Syndrome refers to “impaired physiological function including, but not limited to, metabolic rate, menstrual function, bone health, immunity, protein synthesis, cardiovascular health, caused by relative energy deficiency” [41].

## 5. Conclusions

Eating disorders and disordered eating are serious conditions with complex and multifactorial etiology that affect mental and physical health. Regardless of gender, age, body size, culture, socioeconomic background, or athletic ability, disordered eating can strike any athlete of any sport at any time [8].

Adolescent athletes, as any person with disordered eating, are at increased risk to develop several nutritional and medical problems, which are associated with the use of unhealthy behaviors to change or control their weight. Malnutrition, such as energy and nutrient deficiencies, is expected to develop due to inadequate food consumption, as well as purging behaviors. Adolescent athletes are more exposed to nutritional risks, as their bodies are attempting to balance their increased needs for growth and training, at the same time. Data on the long-term consequences of disordered eating among adolescent athletes are sparse. Nevertheless, it might be the case that some problems linked to their poor nutritional status, such as low bone mineral density, will remain stable during their adult life.

Prevention and early diagnosis of eating disorders and disordered eating is crucial. The “Ad Hoc Research Working Group on Body Composition, Health and Performance”, under the auspices of the IOC Medical Commission in its position statement suggests specific measures in order to decrease the percentage of athletes who use unhealthy weight-loss methods and suffer from disordered eating. Specifically, the suggested measures are: educational programs against extreme dieting and disordered eating, taking into serious consideration the athletes who are trying to lose weight or change their body composition, modification of weight-related regulations in weight-class sports and development of concrete “does not start” (DNS) criteria to protect athlete’s health [14]. Prevention programs should be initiated as early as at 9–11 years of age [14].

Prevention educational programs should focus on adolescent athletes and their parents, as well as their coaches. The programs’ topics may include understanding of physical development during adolescence, information on balanced nutrition for health and performance, healthy ways to control body weight, health and performance effects of unhealthy weight control methods, body composition assessment techniques, avoidance of negative comments and pressure on body weight in the context of sports and family environment, and information on professionals who may advice athletes on nutrition and eating disorders [8,10,14].

Moreover, the regular use of eating disorders (ED) screening tools as a part of the regular medical monitoring could be another useful preventive measure in order to detect eating disorders at an early stage [40]. In case of diagnosis of an eating disorder, the treatment protocol recommended by the treatment team (consisting of a doctor, a psychologist and a dietitian) should be implemented and the training program should be modified according to the athlete’s condition [8,10].

Future research could focus more on long-term effects of disordered eating in athletic populations, as the majority of studies present its direct consequences in health and performance, leaving a knowledge gap regarding the nutritional and health problems that athletes may face after completion of their athletic life. Longitudinal studies on adolescent athletes will be of great importance, especially regarding the consequences that may arise in adult life. Furthermore, there is a need for additional research on specific nutrient deficiencies, as well fluids and electrolytes’ imbalances, in correlation to disordered eating. Use of nutritional intake assessment tools and clinical interviews, in combination with ED screening tools, will be useful for this purpose. Finally, interventional studies in adolescent athletes are also extremely necessary, so that prevention and treatment protocols may be issued and applied in various sports, as it is already happening in college-age and adult athletes.

## Data Availability

Not applicable.

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
