# Peer review of "Nutritional Risks among Adolescent Athletes with Disordered Eating"

_children, 2021, doi:10.3390/children8080715_

Round 1
Reviewer 1 Report
I very much enjoyed reading this interesting and well researched paper – it’s clear the authors are knowledgeable in the subject area and have presented a coherent review of current literature. The key themes and topics covered provide a useful reference for the issues affecting this vulnerable population group.
My main comments refer to the materials and methods presented on page 2; I realise the paper is not presented as a systematic review, but I feel more detail is required to outline how many papers were obtained and reviewed; types of paper reviewed (e.g. primary research, systematic reviews or metanalyses, etc.).
In addition, a clear definition of how the authors define ‘adolescence’ for the purpose of their review needs to be included – what age range and criteria were specified and did this inform the selection of papers for review? On page 2 the authors make reference to the body of literature focussing on adult athletes, ‘mainly of college-age’ – were studies on college students (>18 years?) excluded? Did they only include studies of high school aged (<18years) adolescents? WHO defines adolescence from 10-19 years whilst some propose 10-24 years (https://www.thelancet.com/journals/lanchi/article/PIIS2352-4642(18)30022-1/fulltext )
I think the distinction in age range and what has been included in the review need to be explicit as this will help interpretation of results in some cases (e.g. how representative are studies which cover age-related nutritional outcomes such as LBMD (3.3.)?) and has implications for the diagnosis and prevention strategies outlined on page 9 plus informing future research.
p.6 – final paragraph; do you mean Tae Kwondo?
Finally, an observation about the title – the paper scope of the paper is more than just ‘disordered eating attitudes’. The authors cover patterns of disordered eating, eating behaviours and eating disorders, in addition to attitudes but I feel the title does not reflect the full scope of the paper; in fact ‘attitudes’ are not included in the list of search terms listed in the methods section and therefore I wonder whether the word should be removed from the title? Perhaps “Nutritional Risks among Adolescent Athletes with Disordered Eating” would be sufficient as this encompasses the spectrum of attitudes and behaviours as outlined by the authors in their definition on page 2.
Reviewer 2 Report
The authors presented in the current review an interesting and slightly investigated topic, of fundamental importance for different aspects surrounding the child, adolescent, and youth athletes, and they did a great effort to produce a well written work. Although the quality of the work is appreciable, the work may require some the work requires implementations to provide more detailed considerations and cover aspects that may be of primary importance in the topic
In the present study the authors aimed to review the existing literature on the common nutritional risks and malnutrition issues faced by adolescent athletes with disordered eating attitudes or eating disorders and, to highlight the importance of early nutritional interventions regarding the prevention and treatment of disordered eating in adolescents participating in sports.
Here the comments related to the paper.
Introduction
- What are the possible risk factors leading to body image distortion and rising pressure on youth athletes? It could be worth to mention in the introduction linking with your statements:
e.g.: “Despite their high nutritional requirements, athletes often engage in inappropriate
dietary strategies in their attempt to control their body shape and weight, in order to reach
optimal performance or ideal physical standards for their sport” à why?
- Please consider, and underline, not only acute damages but long-term damages as forcing potential talents to stop completely their career or leaving physiological/metabolic or psychological problems all lifelong.
e.g.: “Disordered eating attitudes may also cause nega-tive effects on sport performance due to low energy availability, excessive loss of fat and lean mass, dehydration, and electrolyte abnormal concentration that may increase risk of illness and injury, cause limitations in training quality and consistency and impairment of recovery after training or injuries” à what about long-term impacts?
Materials and methods
- PubMed/MEDLINE®
- Inclusion and exclusion criteria of the articles were only “adult athletes or non-athletes were excluded” and language?
Results
- How many studies you found? In which categories the studies can be grouped (i.e., x% evaluated low energy availability, y% ...., z%...).
- At the end of each section please consider adding a summary of main remarks resulting from data extraction. For the reader will be easier to find two sentences summarizing the main findings for each subchapter.
- The aim was to underline the prevalence of eating disorders among children/adolescent athletes and its health issues and possible compliances but from the studies have been extracted information mainly on eating behaviors. Please try to extract more information and/or if not available mention it at the end of the section or in the discussion section
Discussion
- – “This means that a great percentage of athletes do not have enough energy, not only to face their training demands, but most importantly for the maintenance of their normal physiological functions. This situation may cause medical complications in several body systems and impair growth and maturation.” Go deeply on this... what are the complication for the male athlete for example? What are the most prevalent complications? How can this affect the life and future of an athlete?
Conclusions
In which direction the research field is going, what is right and what not. What can be the right directions to follow for future research and deeply understand this topic?
Round 2
Reviewer 2 Report
Dear authors, congratulations for the extensive work and thank you for making all the suggested changes. I personally think your review has improved a lot and again, I want to congratulate you on the work done.
I wish you the best of luck in the next steps of the review process.